# Microbial Community Shifts in Tea Plant Rhizosphere under Seawater Stress: Enrichment of Beneficial Taxa

**DOI:** 10.3390/microorganisms12071287

**Published:** 2024-06-25

**Authors:** Xiaohua Zhang, Haozhen Li, Bin Li, Kangkang Song, Yuxue Sha, Ying Liu, Shaolin Dong, Di Wang, Long Yang

**Affiliations:** College of Plant Protection and Agricultural Big-Data Research Center, Shandong Agricultural University, Tai’an 271018, Chinakangkangsong234@163.com (K.S.); 15839772009@163.com (Y.L.);

**Keywords:** seawater stress, rhizosphere, high-throughput sequencing, tea plant

## Abstract

Seawater intrusion has a significant impact on the irrigation quality of agricultural water, thereby posing a threat to plant growth and development. We hypothesized that the rhizosphere of tea plants harbors beneficial microorganisms, which may improve the tolerance of tea plants to seawater stress. This study utilized 16s and ITS techniques to analyze microbial community shifts in the tea plant rhizosphere and non-rhizosphere under seawater stress conditions. The findings suggest that seawater stress leads to a reduction in microbial diversity, although the rhizosphere microbial diversity in stressed soils showed a relatively higher level. Moreover, the rhizosphere of the tea plant under seawater stress exhibited an enrichment of plant growth-promoting rhizobacteria alongside a higher presence of pathogenic fungi. Network analysis revealed that seawater stress resulted in the construction of a more complex and stable rhizosphere microbial network compared to normal conditions. Predictions of bacterial potential functions highlighted a greater diversity of functional groups, enhancing resource utilization efficiency. In general, the rhizosphere microorganisms of tea plants are jointly selected by seawater and the host. The microorganisms closely related to the rhizosphere of tea plants are retained and, at the same time, attract beneficial microorganisms that may alleviate stress. These findings provide new insights into plant responses to saline stress and have significant implications for leveraging vegetation to enhance the resilience of coastal saline soils and contribute to economic progress.

## 1. Introduction

Seawater intrusion disrupts the delicate balance at the interface between fresh groundwater and seawater due to various factors, leading to an inland migration of the freshwater–seawater boundary. This shift results in the salinization of groundwater resources, adversely affecting irrigated agriculture [1,2]. Soil salinity compromises soil quality by depleting nutrients and enzyme activity [3], hindering crop development, limiting agricultural yields, and, in extreme instances, forcing the abandonment of fertile lands, particularly impacting crop production in coastal areas [4,5]. Furthermore, seawater, rich in salt-forming ions such as Cl^−^, NO_3_^−^, SO_4_^2−^, and PO_4_^3−^, can be particularly detrimental to plants with low salt tolerance, manifesting as green discoloration and diminished yield prior to necrosis [6]. Secondary stresses associated with high salinity, such as leaf yellowing, wilting, root necrosis, and compromised antioxidant enzyme activity, can significantly degrade tea quality or lead to plant mortality [7].

Seawater exerts a potent inhibitory effect on various soil microbiological processes, with toxic consequences manifesting within two hours and that can last for at least 48 h [8]. Microorganisms can respond by using osmotic adaptation strategies that involve accumulating low molecular weight organic solutes [9]. However, this adaptive response comes with a significant metabolic cost, often resulting in decreased microbial diversity, except for species that are naturally resilient to changes in salinity [10,11]. Different microbial communities exhibit varying levels of resilience to salt stress, with some showing only minor effects. Elevated Cl^−^ concentrations have been shown to initially suppress nitrification and denitrification, but associated microbial populations seem to adapt to these high ion levels over time [12]. The selection of microorganisms by seawater stress can impact soil biogeochemical cycles, subsequently influencing plant growth and yield over time. For instance, nitrifying bacteria and denitrifying bacteria contribute to soil nitrogen loss [13]. While episodic saltwater intrusions can alter the composition of microbial communities, they may not necessarily impair overall ecosystem function, and these changes are likely to persist [14]. Introducing salt-adapted beneficial microorganisms can boost soil enzyme activities and mitigate the impact of salinity on native microbial groups, thereby promoting improved plant or crop growth [15]. Therefore, it is essential to identify shifts in microbial communities due to seawater stress and acknowledge the vital roles certain microbes play in maintaining community balance.

Plant root systems are surrounded by a diverse microbiome, attracted by specific nutrients, and chosen under specific conditions to form a strong symbiosis with the host plant [16]. Previous studies have indicated that mycorrhizal fungi may work together with phosphorus-solubilizing bacteria to improve plant growth [17,18,19]. On the other hand, higher levels of denitrification and nitrification genes in degraded meadows are associated with an increased risk of soil nitrogen depletion, where highly active microorganisms play a role in accelerating nutrient loss [20]. Khan et al. highlighted the potential of Arthrobacter and Bacillus species in mitigating the negative effects of salt stress on soybeans through the production of indole-3-acetic acid (IAA) and phosphorus solubilization of phosphorus [21]. Recent research has concentrated on isolating strains of plant growth-promoting rhizobacteria (PGPR) from this environment that can enhance peanut growth under salinity stress. These PGPR strains reportedly produce exopolysaccharides, promoting biofilm formation and generating osmoprotectants and antioxidant enzymes. These compounds notably assist plants in overcoming osmotic stress and oxidative damage in salty ecosystems. Additionally, PGPR strains facilitate nutrient uptake and hormone production, further promoting plant growth [22,23]. Plant–microbe interactions offer a novel lens through which to view plant adaptability to environmental fluctuations. The rhizosphere microorganisms associated with tea plants are crucial in mitigating abiotic stress. Research indicates that tea plants utilize microorganisms like *Rhodotorula glutinis*, *Clostridium pasteurianum*, and arbuscular mycorrhizal fungus to enhance resistance against Metal Ion Toxicity and drought [24,25]. Additionally, numerous bacteria with the ability to promote tea plant growth under salt stress have been identified in the rhizosphere of tea plants [26,27,28]. Consequently, comprehending alterations and enrichment patterns within the rhizosphere microbial community in the rhizosphere is crucial for identifying microorganisms that are specifically favored by seawater stress. Utilizing these microorganisms could increase crop yields in saline conditions and contribute to the knowledge for the restoration and conversion of tidal flats.

The rhizosphere of tea plants harbors a unique and highly selective microbial community that is sensitive to climate change [29,30,31]. This phenomenon is progressively impacting agricultural areas, including tea plantations [32]. However, the resilience of tea plant rhizospheric microbiota to seawater intrusion remains poorly understood. Investigating the response of tea plant rhizosphere microbiota to seawater stress can aid in identifying and studying potential microorganisms within the tea plant rhizosphere that can confer resistance to seawater stress. This knowledge can be utilized in various applications to enhance resilience in different contexts. This study aimed to investigate how the tea plant rhizosphere responds to irrigation with seawater-contaminated water compared to that of plants grown under standard conditions. The goal was to determine the impact of seawater on the rhizospheric microbial composition and function of the tea plant rhizosphere and to identify beneficial microbial species that could enhance resistance to seawater stress.

## 2. Materials and Methods

### 2.1. Experimental Design

To figure out the response of tea rhizosphere microbial communities to seawater stress, the seawater irrigation trial was conducted in a tea plantation in China in Rizhao, Shandong Province (35°32′59″ N, 119°36′6″ E), which was used until the beginning of the experiment. The tea plant species selected for this study was the C. sinensis cv. Huangshanzhong with an average age of 2 years. All the teas were grown under the same environmental conditions with the same doses of irrigation and fertilization treatments.

Irrigation water used in seawater stress treatments was contaminated with seawater. The results of seawater quality testing are as follows: 5.06 ms/cm electrical conductivity (EC), which was recognized as brackish water, 632 mg/L hardness, and a pH of 7.87. The pH of the control water was 7.2. The irrigation method follows the traditional use of diffuse irrigation in local tea gardens once a week. Meanwhile, Normal growth populations were sampled in adjacent land, which grew in the nearest tea plantation at distances of <500 m from the seawater irrigated sampling site.

### 2.2. Soil Sampling and Determination of Physicochemical Soil Properties

Following three weeks of seawater stress, a significant portion of tea plants succumbed, while some managed to survive. Subsequently, the seawater stress was prolonged for an additional two weeks to ascertain the adaptability of the surviving tea plant. The normal rhizosphere soil (NRS) and the affected rhizosphere soil (ARS) were then collected for further analysis. The tea plant was carefully excavated by removing soil from a 20 cm^2^ area around the seedlings. The tea plant was shaken vigorously to remove large soil particles, and soil adhering to the root surface was collected from five plants with a brush and sieved through a 2 mm sieve [33]. Normal irrigated (NBS) and seawater irrigated (SBS) plant-free soils were also collected. The soils were divided into five replicates after homogenization to obtain a total of 20 soil samples. Each soil sample was divided into two subsamples; one sub-sample was stored at 4 °C for measurement of soil physico-chemical properties, and the other sub-sample was stored at 80 °C for extraction of rhizosphere microbial DNA within 24 h.

Soil pH and electrical conductivity (EC) were assessed at a 5:1 water-to-soil-mass ratio. Total nitrogen (TN) was quantified using the standard Kjeldahl method, while Skalar Continuous Llow Analyzer 5000 (Skalar Analytical BV, Breda, The Netherlands) was utilized for analyzing available nitrogen (AN) levels. Available phosphorus (AP) content was determined through the acetate-flame photometer method, and organic matter (OM) was measured using the potassium dichromate heating oxidation-volumetric technique.

### 2.3. DNA Extraction, Amplification, and Sequencing

DNA was extracted from nine 0.5 g subsamples (*n* = 20) of each soil sample, as previously described by Ramírez-Villanueva et al. explained [34]. Therefore, 4.5 g of soil DNA can be extracted from each soil sample. The 16S rRNA genes from specific regions (16S V3-V4) were amplified using the primer pairs 341F (5′-CCTAYGGGRBGCASCAG-3′) and 806R (5′-GGACTACNNGGGTATCTAAT-3′) with barcodes [35]. For the ITS region, full-length primers ITS1F-F (5′-CTTGGTCATTTAGAGGAAGTAA-3′) and ITS1F-R (5′-GCTGCGTTCTTCATCGATGC-3′) were utilized [36]. PCR reactions were conducted with 15 µL of Phusion^®^ High-Fidelity PCR Master Mix (New England Biolabs, Ipswich, MA, USA), 2 µM of forward and reverse primers, and approximately 10 ng of template DNA. The thermal cycling protocol included an initial denaturation at 98 °C for 1 min, followed by 30 cycles of denaturation at 98 °C for 10 s, annealing at 50 °C for 30 s, and elongation at 72 °C for 30 s, with a final extension at 72 °C for 5 min.

Subsequently, the PCR products were mixed with an equal volume of loading buffer and subjected to electrophoresis on a 2% agarose gel for visualization. The PCR products were then combined in equimolar ratios and purified using Universal DNA (TianGen, Beijing, China).

Sequencing libraries were prepared using the NEB Next^®^ Ultra DNA Library Prep Kit (Illumina, Foster City, CA, USA) following the manufacturer’s guidelines, including the addition of index codes. The quality of the library was evaluated using an Agilent 5400 Bioanalyzer (Agilent Technologies Co., Ltd., Santa Clara, CA, USA). Finally, the library was sequenced on an Illumina NovaSeq platform, generating 250 bp paired-end reads. The sequences were submitted to the NCBI Sequence Read Archive under the identification PRJNA1121506. The overall coverage for all samples exceeded 97%, and each sample’s rarefaction curve appeared level (Appendix A). This suggested that the sequencing depth was sufficient and capable of capturing the majority of microbial diversity information present in the samples.

### 2.4. Statistical and Bioinformatics Analysis

The examination was carried out by following the “Atacama soil microbiome tutorial” provided by Qiime2docs and utilizing customized program scripts (https://docs.qiime2.org accessed 1 October 2022). In summary, the initial step involved importing raw data FASTQ files into a format compatible with the QIIME2 system using the qiime tools import program. Subsequently, the sequences from each sample underwent demultiplexing. Use the QIIME2 commands “qiime cutadapt trim-paired” and convert to visualizable qzv files for quality filtering, where the middle of box quality score for bacteria and fungi is >35, indicating suitable sequencing quality. Use the QIIME2 command “qiime dada2 denoise-paired” to trim, de-noise, merge, and eliminate chimeric sequences for bacteria (--p-trim-left-f 23 --p-trim-left-r 26 --p-trunc-len-f 249 --p-trunc-len-r 249) and fungi (--p-trim-left-f 28 --p-trim-left-r 26 --p-trunc-len-f 249 --p-trunc-len-r 246) using the QIIME2 dada2 plugin to produce the feature table of amplicon sequence variant (ASV) (Appendix A) [37]. Based on table.qzv, decide on standardized sampling depth, aiming for a high value while retaining all samples. Following this, the QIIME2 feature-classifier plugin was employed to align ASV sequences with pre-trained GREENGENES (v13-8) and UNITE (v8.2) databases for the generation of the taxonomy table [38]. To eliminate any potentially contaminating mitochondrial and chloroplast sequences, the QIIME2 feature table plugin was used. Single-factor analysis of variance was performed using Duncan’s multiple range test (*p* < 0.05) in SPSS 24.0 (SPSS Inc., Chicago, IL, USA) to analyze the soil environmental parameters.

Statistically significant group variations were assessed using analysis of variance (ANOVA) or Wilcoxon, with adjustment for false discovery rate (FDR) in *p* values. The relationship between soil properties and α-diversity was analyzed using the “corrplot” package in R version 3.6.3 [39]. To examine differences in community composition, Principal coordinate analysis (PCoA) and permutational multivariate analyses of variance (PERMANOVA) were conducted based on the Bray–Curtis distance utilizing the “labdsv” and “vegan” packages [40,41]. Bacterial and fungal genera with the top 50 abundances were selected, and Z-score normalization was performed within each replicate. A heatmap was then generated using the “pheatmap” package. To investigate the enrichment patterns of tea rhizosphere microorganisms at the ASV and genus levels in various treatment plots, the trimmed mean of M values (TMM) method was utilized to normalize bacterial and fungal ASVs, followed by a differential abundance analysis using the generalized linear model (GLM) approach with the ‘EdgeR’ package [42,43]. In order to discover representative microorganisms in the rhizosphere of the different treatments, linear discriminant analysis effect size (LEfSe) was utilized to detect significantly abundant bacterial and fungal taxa using a standardized scaling factor of 1,000,000 [44]. The analysis of microorganisms was carried out using the Wekemo Cloud Platform (https://bioincloud.tech accessed on 1 October 2023).

Co-occurrence network analysis was performed on relative abundances at the genus level to assess interactions between microbial communities. Data were filtered prior to the construction of the network to remove rare genera with relative abundance less than 0.01% to avoid null values that could lead to spurious correlations. Vegan and psych packages in R were used to perform network analysis. The correlation matrix was performed by Spearman rank correlation (ρ > 0.7, *p* < 0.05) between microbial taxa, and *p* values were adjusted by multiple testing correction using the Benjamini–Hochberg method to minimize the chances of obtaining false-positive results [45]. The network was visualized using the Frucherman Reingold layout in Gephi (v0.10.1) software [46]. Gephi was used to calculate the degree and betweenness centrality and other topological indexes of the network. The genus with the highest degree and highest closeness centrality of the top 10 was defined as the hub taxa.

Functional profiles of bacteria were predicted by utilizing the Functional Annotation of Prokaryotic Taxa (FAPROTAX) tool (function decoupling) [47]. The STAMP software was utilized to conduct Welch’s one-sided *t*-tests, aiming to identify functions that exhibited enrichment in the rhizosphere [48]. In order to ecologically and functionally classify fungi, the FUNGuild (Fungi Functional Guild) V1.0 online platform was employed [49]. Subsequently, the obtained Amplicon Sequence Variants (ASVs) from high-throughput sequencing were submitted to the FUNGuild platform for analysis. The outcomes were then retrieved to analyze fungal communities and establish connections between fungal species classification and functional guilds using bioinformatics techniques (fungal community composition).

## 3. Results

### 3.1. Effects of Seawater Stress on the Content of Soil Chemical Properties

The soil chemical analyses indicated that seawater stress had an impact on the chemical composition of both the rhizosphere and bulk soil (Table 1). Seawater irrigation led to a decrease in pH, AN, EC, TN, TC, and SOM in the rhizosphere, as well as pH, TN, and TC in the bulk soil (*p* < 0.05). However, there was an increase in AP content. In the control site, the rhizosphere generally showed higher chemical properties compared to the bulk soil, except for pH and TC. After rhizosphere soil was exposed to seawater stress, only EC and SOM levels increased significantly (*p* < 0.05). On the other hand, pH, TN, TC, and SOM concentrations decreased due to seawater exposure, while AP content increased. Notably, the AP changes in the SBS sample were particularly notable, with the highest value of 22.86 mg/kg—71.08% and 38.8% greater than those in the NBS and ARS, respectively (*p* < 0.05). However, there were no significant differences in AP content within the rhizospheres between the two treatment groups. Overall, the significant impact of seawater stress on soil chemistry indicates potential shifts in soil fertility and nutrient availability under such conditions.

### 3.2. Effects of Seawater Stress on Microbial Community Structure and Diversity

The Shannon index demonstrated variability among the samples, with NBS > NRS > ARS > SBS (Figure 1A). Seawater stress caused a significant decrease in the bacterial Shannon index (*p* < 0.05), with a more noticeable decline in bulk soil compared to the rhizosphere. Interestingly, the Shannon diversity index was notably lower in the rhizosphere than in the bulk soil at the control site (*p* < 0.05). Conversely, a higher alpha diversity was observed in the rhizosphere of tea plants in plots exposed to seawater stress. The trends in bacterial species richness were consistent with those of the Shannon index. In terms of soil fungi, the Shannon diversity index peaked in ARS (6.66) and was lowest in NRS (5.35) (*p* < 0.05), suggesting an increase in specific taxa and restructuring of fungal communities under seawater stress (Figure 1B).

Principal coordinate analysis (PCoA) using Bray–Curtis distances effectively separated the bacterial and fungal sampling sites on the first principal coordinate, with distinctions between rhizosphere and bulk soil observed on the second principal coordinate (Figure 1C,D). Results from permutational multivariate analysis of variance (PERMANOVA) revealed that seawater stress had a significant impact on fungal communities in both bulk and rhizosphere soils (R^2^ = 0.76 for bulk soil, R^2^ = 0.78 for rhizosphere soil) (Appendix A). Furthermore, bacterial communities in samples with and without tea plants showed varying levels of similarity, with ARS and NRS samples displaying greater similarity (R^2^ = 0.48) compared to SBS and NBS samples (R^2^ = 0.65). The dominant bacterial genera observed were Candidatus Koribacter, Candidatus Solibacter, and Rhodoplanes, representing 17.22%, 10.84%, and 12.04% of the total sequences, respectively (Figure 1E). Among the fungal genera, Mortierella, Pseudeurotium, and Saitozyma were the most abundant, accounting for 24.53%, 9.11%, and 6.45% of the total sequences, respectively (Figure 1F). Interestingly, the presence of Rhodoplanes was significantly higher in seawater-stressed samples, while Candidatus Koribacter and Candidatus Solibacter were notably reduced (Appendix A). At the genus level within fungi, Mortierella exhibited a 47.6% decrease, accompanied by significant increases in Pseudeurotium, Saitozyma, and Trichoderma (Figure 1F). In brief, both seawater stress and tea plants significantly affected both the diversity and composition of the microbial community.

### 3.3. Selective Enrichment of Microbial Genera in Tea Plant Rhizospheres Exposed to Seawater

Cluster analysis was conducted on the top 50 genera and visualized through heat maps (Figure 2A,B). The clustering of relatively abundant bacterial and fungal taxa demonstrated a clear separation between treatments, indicating that seawater stress resulted in a decrease in the abundance of most microbial taxa. The differential abundance testing assessed the species filtration and selection processes from bulk soil to rhizosphere. At the ASV level, a significant reduction in the number of distinct ASVs post-stress was observed, suggesting that seawater stress diminished the selection pressure exerted by the tea plant (Appendix A). Conversely, there was an increase in the number of fungal ASVs associated with the tea plant rhizosphere. A total of 36 bacterial genera and 43 fungal genera displayed significant variation in NRS, while 13 bacterial genera and 51 fungal genera exhibited notable variation in ARS (Figure 2C,D). Genus-level co-enrichment in both NRS and ARS included *Corynebacterium*, *Oscillospira*, *Tetragenococcus*, and *Staphylococcus*. *Actinomycetospora* and *Kocuria* were specific to ARS. Only 17 percent of the enriched fungal genera overlapped with those enriched in ARS.

LEfSe analysis allowed for a finer taxonomic comparison among treatments at the ASV level (Figure 3A,B). An analysis of the top ten rhizosphere biomarkers was conducted (Appendix A). In the NRS, bacteria from the phyla Pseudomonadota (*Rhodoplanes*, *Reyranella*, *Steroidobacter*, *Phenylobacterium*, *Sphingomonas*, *Dokdonella*, and *Desulfovibrio*), Bacteroidota (*Niastella*), and Acidobacteriota (*Edaphobacter*) were more prevalent. ARS exhibited greater phyla diversity, hosting Pseudomonadota (*Ralstonia*, *Rhodoblastus*, *Devosia*, and *Methylobacterium*), Bacillota (*Bacillus* and *Ammoniphilus*), Actinomycetota (*Convolvulus* and *Frankia*), Bacteroidota (*Mucilaginibacter*), and Mycoplasmatota (*Asteroleplasma*). Contrasting with bacterial patterns, fungi, particularly Ascomycota genera (*Cyphellophora*, *Gibberella*, *Penicillium*, and *Sarocladium*), flourished following seawater stress. Collectively, these results indicate that while plant presence led to the enrichment of similar bacterial genera across different rhizospheres, seawater stress induced the recruitment of specific bacterial taxa at distinct phylum levels. Despite the overarching impact of seawater, the rhizosphere environment supported the proliferation of specific fungal phyla.

### 3.4. Network Complexity of Rhizosphere Microbiota Influenced by Seawater Stress

Microbial co-occurrence network analysis was utilized to visualize the symbiotic relationships among the rhizosphere microbial community of tea plants under seawater and non-seawater treatments (Figure 4A,B). The ARS treatment exhibited the most interconnected microbial network, with a higher average degree compared to the NRS treatment. Moreover, the ARS treatment had a greater number of network edges (5032 in bacteria, 6134 in fungi) compared to the NRS treatment (4706 in bacteria, 5500 in fungi). While most correlations in all networks were positive, the bacterial microbial network within the ARS treatment showed 36% negative interactions, which was higher than the 31% observed in the NRS treatment (Table 2).

The identification of potential ‘hub taxa’ was based on the top 10 nodes with degree values and closeness centrality in the microbial networks (Appendix A). In the fungal networks, hubs mainly belonged to the Ascomycota phylum, while in the bacterial networks, hubs from the ARS treatment showed more phylum diversity compared to the NRS treatment, which was dominated by Proteobacteria. Notably, the genera *Mesorhizobium* and *Achromobacter* within Proteobacteria were identified as crucial network hubs in the rhizosphere. In conclusion, the rhizosphere network exhibits greater complexity and stability under seawater stress, with potential interactions between beneficial microorganisms.

### 3.5. Functional Shifts in Rhizosphere Microbiome Due to Seawater Stress

Functional annotation of prokaryotic taxa (FAPROTAX) in rhizosphere and bulk soil samples from normal and stressed sites revealed a higher abundance of significantly enriched rhizosphere-specific functional groups at the stressed site compared to the normal site (Wilcox test, *p* < 0.05, Figure 5A). In total, 14 functional groups were found to be enriched in the rhizosphere across different treatments, with a focus on nitrogen cycling. Notably, functions such as aerobic chemoheterotrophy, anoxygenic photoautotrophy, general chemoheterotrophy, hydrocarbon degradation, methanol oxidation, methanotrophy, and ureolysis were enhanced in the stressed rhizosphere (Figure 5B).

Fungal functional classification and corresponding abundance levels across various treatment samples were determined using FUNGuild function prediction, which classifies them as pathotrophs, symbiotrophs, or saprotrophs (Table 3). The results showed that saprotrophic, Saprotroph–Symbiotroph, and Pathotroph–Saprotroph fungi were the predominant functional groups in all sample sets (Table 3), consistent with the increase in relative abundance of Ascomycota after seawater stress (Figure 2B). Additionally, the fungal functional profile in ARS revealed a notable presence of Pathotroph–Symbiotroph and Saprotroph groups, indicating significant changes in fungal communities toward utilizing host damage for nutrients in stressful conditions. Generally, stressed tea plant rhizosphere microorganisms possessed more functional redundancy.

## 4. Discussion

### 4.1. The Microbial Diversity of Tea Rhizosphere Presents More Stable under Seawater Stress

An increase in soil salinity disrupts the osmotic balance of microbial cells, ultimately causing their death [50]. This research demonstrates that saline stress significantly reduces microbial diversity. Analyses using the Bray–Curtis difference matrix indicated that the groups could be separated based on the treatment. Although salinity stress generally reduced microbial diversity in both rhizosphere and bulk soil, notably, the rhizosphere microbial diversity in stressed soils exhibited a relatively higher level. This is due to the fact that specific rhizosphere microbial communities have been shaped by thousands of years of plant evolution and are able to assist the host plant under biotic and abiotic environmental stresses [51]. Plants significantly affect the diversity of rhizosphere microbial communities by selectively recruiting symbiotically associated microorganisms. This selectivity often leads to varying impacts on the composition, distribution, and diversity of these microbial communities within the rhizosphere [52,53]. This explained that the diversity of the rhizosphere microbial community was relatively low under normal conditions and maintained at a high level compared to the bulk soil under seawater stress in this study. There are reports indicating that the tea rhizosphere flora demonstrates a high level of selectivity, resulting in a reduction in the diversity of rhizosphere microorganisms and the enrichment of particular microorganisms [31]. However, further research is needed to determine if the selected microbial community in the tea rhizosphere can enhance its ability to withstand saltwater stress. Additionally, the results of beta diversity revealed that the bacterial communities of ARS and NRS exhibited greater similarity, whereas the fungal community structures differed significantly. This difference may be attributed to the substantial increase in fungal diversity under seawater stress, and new fungi that can adapt to seawater occupy the previous niche. In addition to the protective effect of the special chitin cell wall structure of the fungus itself [54,55], the organic acids in the tea plant root exudates can contribute to the cultivation process. By promoting soil acidification, this unique property of the tea plant also enhances the proliferation of fungi [56]. Field experiments have shown that the natural soil acidification rate of 0.071 units per year [57,58] constructs an environment conducive to the growth of fungi. Research findings indicate that seawater stress has a significant impact on the microbial community, leading to a reduction in the diversity of rhizosphere microorganisms. Nevertheless, the tea plant rhizosphere also attracts certain microorganisms that are capable of thriving in both seawater and rhizosphere conditions. It is hypothesized that alterations in diversity, possibly driven by fungi, highlight the crucial role of tea plant rhizosphere exudates in shaping microbial communities. Plants and stress are not the only factors that influence microorganisms. Various environmental factors play a role in shaping microbial communities, particularly in soil. Future research could investigate how seawater stress impacts the physical and chemical properties of soil, in order to better understand the mechanisms driving changes in microbial communities.

### 4.2. The Rhizosphere Microorganisms of Tea Plant Are Screened by Seawater and Host

Studies of microbial diversity have confirmed that the screening and recruitment of rhizosphere microorganisms were influenced by both seawater stress and host plants. Seawater stress significantly affects the microbial community. Some microorganisms with high adaptability, such as *Rhodoplanes*, *Trichoderma,* and *Pseudeurotium*, can survive in extreme environments and gradually adapt to the soil environment under seawater stress. Prior research has revealed the presence of *Kocuria* in halophytes [59,60]. This genus comprises various halotolerant strains, such as *K. rhizophila*, *K. marina,* and *K. kristinae,* capable of thriving in environments with different concentrations of NaCl [61]. *Kocuria* was found to be significantly more abundant in the rhizosphere of tea plants when exposed to seawater stress. This enrichment is not solely attributed to the impact of seawater stress; plants also play a role in selecting microorganisms, with host plants able to recruit beneficial microorganisms to support plant growth in stressful conditions [62,63]. Studies have established connections between *Actinobacteria*, *Kocuria*, and seawater stress in the soil surrounding tea plants, indicating the potential of these microorganisms to enhance plant growth in saline conditions [64,65]. Studies show that a strain of *K. rhizophila* isolated from the maize rhizosphere produces IAA, thus enhancing maize’s salt tolerance upon inoculation [66]. Furthermore, halophytic plants cultivated in seawater media exhibited a prevalence of *Kocuria* among the isolated salt-tolerant bacteria, and its bacterial isolates were shown to positively influence barley seed germination and seedling growth [67]. This study observed a significant enrichment of *Kocuria* in the tea plant rhizosphere under seawater stress. Subsequent LEfSe analysis indicated that tea seedlings under seawater stress recruited a variety of beneficial microbes, such as *Devosia*, *Bacillus*, *Frankia*, *Penicillium*, and *Sarocladium*, all recognized as plant growth-promoting rhizobacteria [68].

In both the normal and seawater treatment groups, there was a significant overlap in the microorganisms enriched in the rhizosphere, indicating that the tea plant enrichment strategy is equally effective in enhancing rhizosphere microorganisms regardless of seawater stress. Future studies could explore these microorganisms that consistently inhabit the rhizosphere of tea plants, as they may play a crucial role in maintaining the balance and promoting normal growth of plant rhizosphere. These consistently existing microorganisms can be considered as the core, while other specifically recruited microorganisms act as extensions of this core. These extensions could potentially offer tea plants increased survival opportunities under seawater stress. In addition, as mentioned above, metabolism shapes the microbial community. The emergence of these beneficial microorganisms may also be related to the root metabolites of tea plants. Future studies should prioritize these microorganisms as key research subjects, investigating their connections with specific metabolites and their involvement in unique metabolic pathways that aid plants in resisting salt stress.

### 4.3. The Rhizosphere Microorganisms of Tea Plant Form Closely Linked under Seawater Stress

The high abundance of certain selected microorganisms may be attributed to their adaptation to environmental changes. Conversely, microorganisms with lower abundance could still have significant roles within the microbial community. The use of co-occurrence patterns to investigate the intricate networks of interactions and the ecological guidelines that dictate the assembly of microbial communities in specific ecological niches has become increasingly common [69]. In the study of the impact of seawater stress on the bacterial and fungal communities of tea plants, the co-occurrence networks in the rhizosphere of tea plants exposed to seawater stress were found to be more complex. The predominance of positive connections among interacting members implies a similarity in responses to environmental fluctuations, leading to synchronous oscillations that render the microbial network unstable [46]. The percentage of negative correlations increased from 31% to 36% in ARS as compared to the NRS network. This indicates that seawater stress triggers competitive interactions among bacteria, enhancing the microbial community’s stability by diminishing the instability that cooperation can introduce [70,71]. The diversity of the ARS sample group decreases, accompanied by the generation of a more complex network, which means that some microorganisms enriched in the rhizosphere of tea plants assume more functional roles and a microorganism overlaps in different functions. These microbes may have facilitated more frequent material and energy exchanges, leading to the formation of tighter networks that help enhance the overall stability and resilience of the ecosystem. In addition, the hub taxa *Mesorhizobium* and *Achromobacter* have been confirmed to be available in the rhizosphere and to form mutualistic relationships with plants, coordinating and participating in plant adaptation to salt stress [72,73]. These microorganisms have multiple functions, including nitrogen fixation, production of plant growth hormones, and symbiosis with plants to enhance stress resistance. This explains the complex network formed by ARS under low diversity. Future research should further investigate the roles of these important microbes in assisting plants to withstand seawater stress. This study only provides a snapshot of the microbial community changes at a specific time point, resulting in a weak understanding of the relationship between microbial communities. To enhance this connection, future research could delve into their shared and distinct characteristics across spatial and temporal scales.

### 4.4. Functional Prediction Revealed the Conservation and Diversity of Tea Rhizosphere Community

Functional annotation of the prokaryotic taxa (FAPROTAX) is a valuable tool for predicting the function of microbial communities using the available literature data. It is important to note that the predictive ability of FAPROTAX may be influenced by biases when there is limited literature on a specific function or when the function of a particular microbial community has not been thoroughly studied. Despite these limitations, FAPROTAX has been widely acknowledged and utilized in various studies [74]. The results revealed that the rhizosphere enrichment functions observed under normal conditions were maintained even under seawater stress. Furthermore, seven new potential functional groups were found to be enriched in the rhizosphere when subjected to seawater stress. When investigating the predicted functions of bacterial communities, it was observed that a significant portion of these functions were associated with nitrogen cycling. This indicates that soil microorganisms play a crucial role in the transformation and uptake of nutrients. Functional groups such as methanol oxidation, methanotrophy, and ureolysis were enriched in the rhizosphere under seawater stress. The symbiotic relationship between methanotrophic and methanol-oxidizing bacteria, along with ureolytic organisms in the plant rhizosphere, plays a crucial role in promoting nutrient cycling and maintaining plant health. Methanotrophs harness atmospheric methane as a carbon source, directly supporting plant growth by contributing to the carbon budget within the rhizosphere [75]. Concurrently, methanol-oxidizing bacteria facilitate the conversion of methanol, a by-product of plant metabolism, into biomass and energy, thereby maintaining a clean and balanced rhizospheric environment conducive to plant health [76]. Furthermore, the presence of ureolytic microorganisms in the rhizosphere enhances nitrogen availability through the hydrolysis of urea into usable forms for plant uptake, which is pivotal for plant growth, particularly in nutrient-limited conditions [77]. Concurrently, pathogenic fungi constitute a significant portion (Table 3), which may be attributed to an array of pathogens exploiting the nutrients produced by root exudates and decomposition for growth, thereby filling the ecological niches of microbes affected by seawater. Collectively, these microbial processes are integral to the rhizospheric ecosystem, optimizing resource efficiency and bolstering plant resilience under seawater stress. Our results further support the core-extension model previously discussed. Even when subjected to seawater stress, all rhizosphere-specific microbial functional groups remain intact and exhibit additional functions that contribute to plant growth and provide more nutrients for microorganisms.

## 5. Conclusions

This study examines the impact of seawater stress on the bacterial and fungal communities in the rhizosphere of tea plants. The research focuses on analyzing microbial differences at the genus level under seawater stress compared to normal conditions, as well as the enrichment status of microbial functions. The findings indicate that seawater stress leads to a reduction in microbial diversity and alters the composition of microbial communities. The rhizosphere microorganisms of the tea plant face dual selective pressures from seawater and the host, resulting in specific core microorganisms that support the plant’s basic functions and growth. Additionally, there are extended microorganisms that assist in resisting seawater stress and enhancing the tea plant’s overall stress resistance. These microorganisms create intricate and resilient networks, showing closer interconnections. Moreover, the microorganisms involved in mitigating seawater stress in tea plants can aid in the plants’ adaptation to such stress by improving resource utilization efficiency. This research could offer valuable insights for improving agricultural practices in coastal areas facing seawater intrusion.

## Figures and Tables

**Figure 1 microorganisms-12-01287-f001:**
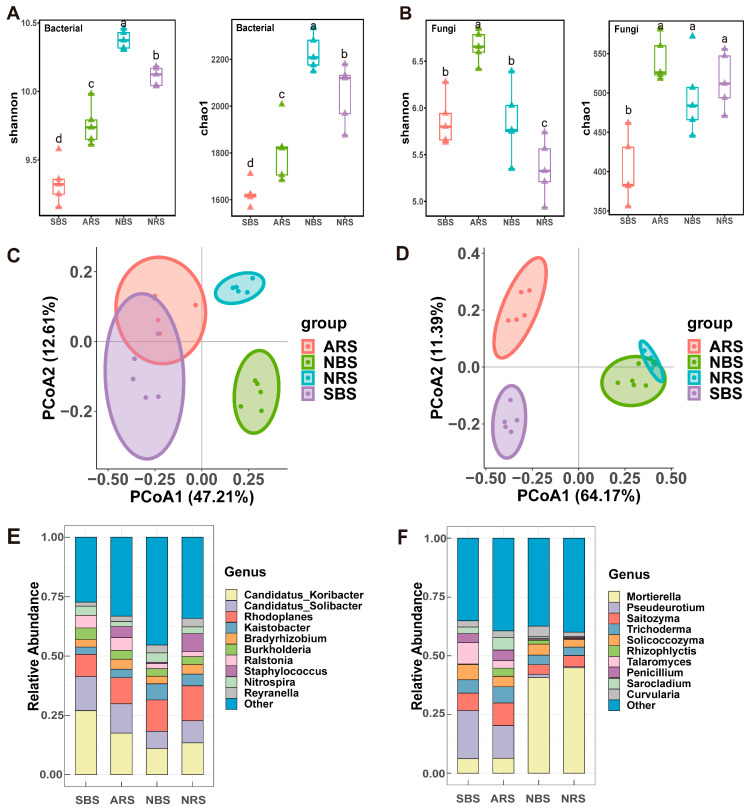
Structure and diversity of the microbial community. (**A**) The Shannon and Chao1 index of bacterial. (**B**) The Shannon and Chao1 index of fungi. Different lower-case letters refer to significant differences between each site based on Wilcoxon test (*p* < 0.05). The PCoA analysis of bacterial (**C**) and fungal (**D**) communities based on Bray–Curtis distances. Average relative abundance of bacterial (**E**) and fungi (**F**) at the genus level. NRS: normal rhizosphere soil. ARS: affected rhizosphere soil. NBS: normal bulk soil. SBS: stressed bulk soil.

**Figure 2 microorganisms-12-01287-f002:**
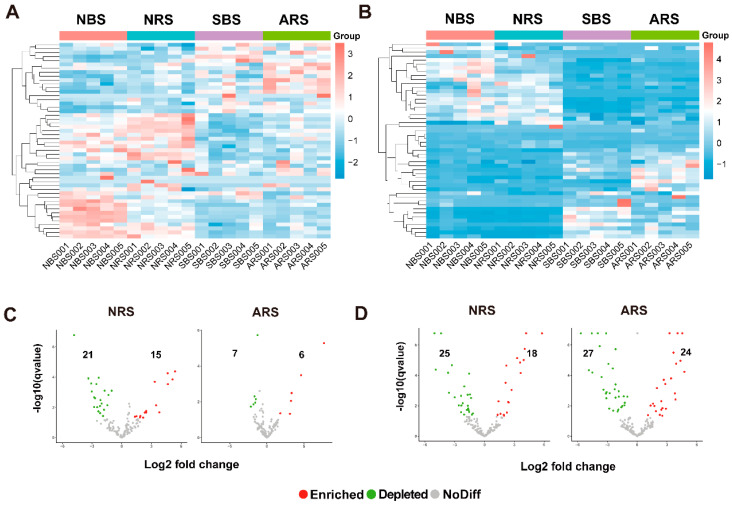
The impact of seawater stress on microbial distribution at the genus level. (**A**,**B**) Combined heat level map of the community composition with cluster analysis. Taxa are clustered according to the degree of similarity distributed among different samples and arranged in a vertical order according to the clustering results. The data were standardized using the Z-score method. Red represents the more abundant genera in the corresponding sample, and blue represents the less abundant genera. The volcano plot illustrating the enrichment and depletion patterns of rhizosphere bacterial (**C**) and fungal (**D**) microbiomes in the control and seawater-stressed samples. Each point represents a genus. Each red point represents an individual enriched genus, and a green point represents an individual depleted genus. NRS: normal rhizosphere soil. ARS: affected rhizosphere soil. NBS: normal bulk soil. SBS: stressed bulk soil.

**Figure 3 microorganisms-12-01287-f003:**
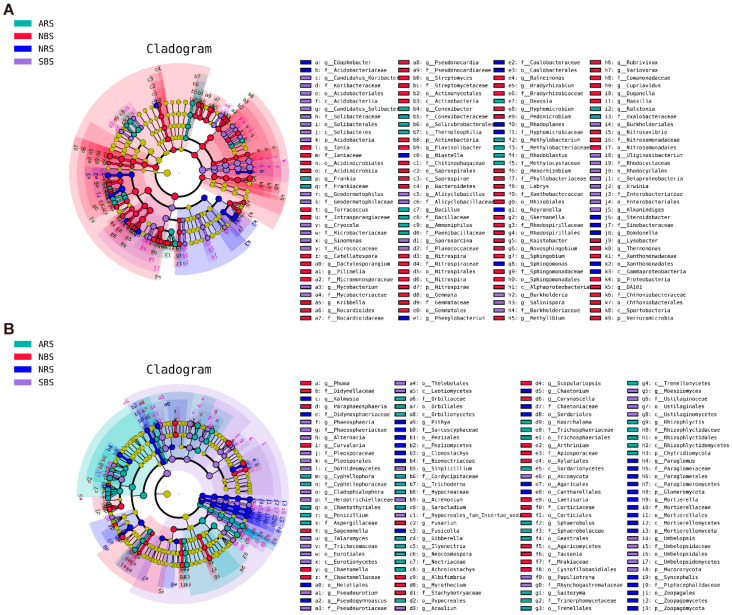
LDA effect size phylogenetic cladogram of bacteria (**A**) and fungal (**B**). Taxa with significantly different successions are represented by colored dots, and Cladogram circles represent phylogenetic taxa from phylum to genus. Only the LDA score > 3 for bacteria and >3.5 for fungi are shown. NRS: normal rhizosphere soil. ARS: affected rhizosphere soil. NBS: normal bulk soil. SBS: stressed bulk soil.

**Figure 4 microorganisms-12-01287-f004:**
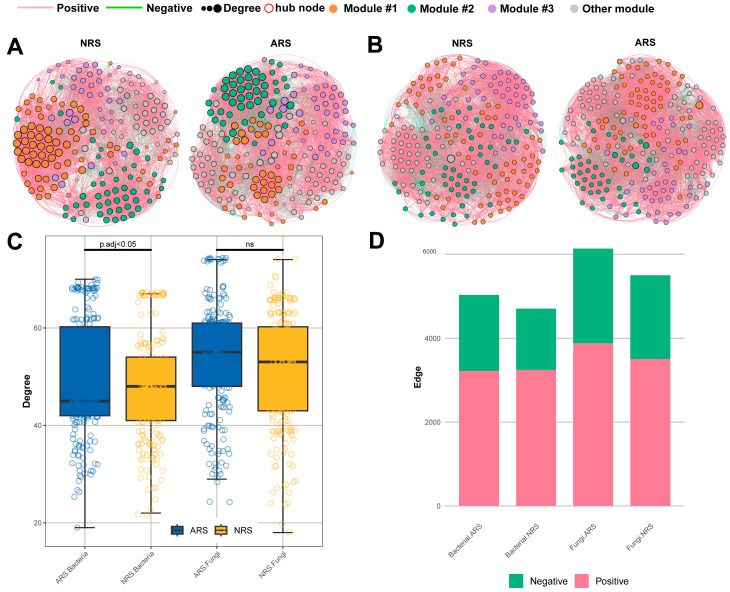
Microbial co-occurrence networks in the rhizosphere. Bacterial (**A**) and fungi (**B**) co-occurrence networks in the rhizosphere. Only compositionality-robust (|*p*| > 0.7) and statistically significant (q < 0.05) correlations are shown. Degree of nodes (**C**) and number of correlations (**D**) in the rhizosphere networks. NRS: normal rhizosphere soil. ARS: affected rhizosphere soil. NBS: normal bulk soil. SBS: stressed bulk soil.

**Figure 5 microorganisms-12-01287-f005:**
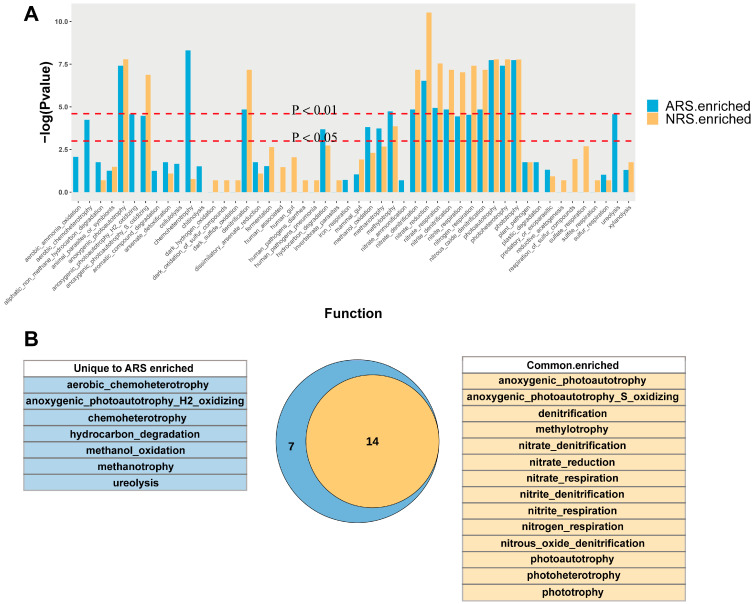
Predicted functional analysis of the tea plant rhizosphere microbiome. (**A**) Functional predictive analysis of significant enrichment in the rhizosphere using FAPROTAX, respectively. Significant differences between bulk soil and rhizosphere soil were compared using Wilcoxon’s test. (**B**) Venn diagram of co-annotated and unique functions that were significantly enriched in the rhizosphere of ARS and NRS. NRS: normal rhizosphere soil. ARS: affected rhizosphere soil. NBS: normal bulk soil. SBS: stressed bulk soil.

**Table 1 microorganisms-12-01287-t001:** Chemical properties of the examined soil.

Group	PH	AN (mg/kg)	EC (μs/cm)	TN (mg/kg)	TC (%)	AP (mg/kg)	SOM (%)
SBS	5.9 ± 0.04 c	30 ± 2.98 ab	40.07 ± 0.64 c	0.82 ± 0.01 c	1.7 ± 0.01 d	22.86 ± 1.29 a	0.58 ± 0.01 c
ARS	5.41 ± 0.05 d	27 ± 4.71 b	113.07 ± 1.9 b	0.8 ± 0.01 c	1.72 ± 0 c	13.99 ± 1.19 b	0.93 ± 0.1 b
NBS	6.59 ± 0.13 a	36 ± 9.25 ab	39.33 ± 0.47 c	0.89 ± 0.03 b	1.76 ± 0.01 a	6.61 ± 0.25 c	0.66 ± 0.03 c
NRS	6.04 ± 0.03 b	42.09 ± 6.57 a	171.73 ± 2.24 a	1.01 ± 0 a	1.74 ± 0.02 b	13.05 ± 2.05 b	1.24 ± 0.14 a

PH: pondus hydrogenii; AN: available nitrogen; EC: electrical conductivity; TN: total nitrogen; TC: total carbon; AP: available phosphorus; SOM: soil organic matter. Data are presented as means ± SEs (*n* = 5). Different letters in the line indicate significant differences (*p* < 0.05). NRS: normal rhizosphere soil. ARS: affected rhizosphere soil. NBS: normal bulk soil. SBS: stressed bulk soil.

**Table 2 microorganisms-12-01287-t002:** Bacterial co-occurrence network characteristics in sample.

		Node	Edge	Positive Edge ^1^	Negative Edge ^2^	Average Degree ^3^	Modularity ^4^	Average Clustering Coefficient ^5^	Average Path Distance ^6^
Bacteria	ARS	204	5032	3219	1813	49.333	1.654	0.582	1.936
	NRS	197	4706	3244	1462	47.77	1.142	0.61	1.94
Fungi	ARS	227	6134	3883	2251	54.004	1.694	0.56	1.952
	NRS	216	5500	3505	1995	50.92	1.68	0.571	1.948

NRS: normal rhizosphere soil. ARS: affected rhizosphere soil. NBS: normal bulk soil. SBS: stressed bulk soil. ^1^: a favorable or beneficial relationship between two nodes. ^2^: an unfavorable or adverse relationship between two nodes. ^3^: the average number of connections each node has. ^4^: degree of nodes tending to differentiate into different network modules. ^5^: measure of the degree to which nodes in a network tend to cluster together. ^6^: average path distance: the average length of the shortest paths between all pairs of nodes in the network.

**Table 3 microorganisms-12-01287-t003:** Changes of predicted soil fungal functional groups. Different lower-case letters refer to significant differences between each site based on Wilcoxon test (*p* < 0.05).

Fungal Functional Groups	SBS	ARS	NBS	NRS
Animal Pathogen–Endophyte–Epiphyte–Fungal Parasite–Plant Pathogen–Wood Saprotroph	0.046 ± 0.002 b	0.068 ± 0.004 a	0.036 ± 0.002 b	0.032 ± 0.001 b
Animal Pathogen–Endophyte–Fungal Parasite–Lichen Parasite–Plant Pathogen–Wood Saprotroph	0.108 ± 0.002 b	0.127 ± 0.002 b	0.117 ± 0.006 b	0.154 ± 0.003 a
Animal Pathogen–Endophyte–Fungal Parasite–Plant Pathogen–Wood Saprotroph	0.037 ± 0.001 a	0.012 ± 0.002 b	0.011 ± 0.001 b	0.005 b
Animal Pathogen–Endophyte–Lichen Parasite–Plant Pathogen–Soil Saprotroph–Wood Saprotroph	0.001 d	0.003 c	0.008 a	0.005 b
Animal Pathogen–Endophyte–Plant Pathogen–Undefined Saprotroph	0.007 ± 0.001 a	0.008 ± 0.001 a	0.005 ab	0.002 b
Animal Pathogen–Soil Saprotroph	0.002 b	0.002 b	0.005 ± 0.001 ab	0.008 ± 0.001 a
Animal Pathogen–Undefined Saprotroph	0.009 ± 0.001 b	0.035 ± 0.001 a	0.004 c	0.002 c
Arbuscular Mycorrhizal	0.025 ± 0.001 b	0.036 ± 0.002 ab	0.007 ± 0.001 c	0.037 ± 0.002 a
Dung Saprotroph–Undefined Saprotroph–Wood Saprotroph	0.032 ± 0.004 a	0.045 ± 0.002 a	0.004 b	0.004 b
Endophyte–Dung Saprotroph–Lichen Parasite–Litter Saprotroph–Plant Pathogen–Soil Saprotroph–Wood Saprotroph	0 c	0.001 c	0.011 ± 0.001 a	0.003 b
Endophyte–Litter Saprotroph–Soil Saprotroph–Undefined Saprotroph	0.266 ± 0.012 b	0.149 ± 0.006 c	0.487 ± 0.007 a	0.486 ± 0.011 a
Endophyte–Plant Pathogen	0.006 ± 0.001 b	0.009 ± 0.001 a	0.001 c	0.002 c
Fungal Parasite	0 b	0.002 ± 0.001 b	0.023 ± 0.005 a	0.027 ± 0.001 a
Fungal Parasite–Soil Saprotroph–Undefined Saprotroph–Wood Saprotroph	0.002 c	0.004 c	0.042 ± 0.001 a	0.019 ± 0.001 b
Soil Saprotroph–Undefined Saprotroph	0.012 ± 0.002 b	0.019 ± 0.001 a	0.001 c	0.001 c
Undefined Saprotroph	0.321 ± 0.004 a	0.355 ± 0.014 a	0.142 ± 0.006 b	0.106 ± 0.002 b

NRS: normal rhizosphere soil. ARS: affected rhizosphere soil. NBS: normal bulk soil. SBS: stressed bulk soil.

## Data Availability

The original contributions presented in the study are included in the article and Appendix A, further inquiries can be directed to the corresponding authors.

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
