# Peer review of "Microbial Community Shifts in Tea Plant Rhizosphere under Seawater Stress: Enrichment of Beneficial Taxa"

_microorganisms, 2024, doi:10.3390/microorganisms12071287_

Round 1

Reviewer 1 Report

Comments and Suggestions for Authors

This manuscript describes a study characterizing the effects of sea water contamination of the rhizosphere communities of tea plants in China. While the study is well presented and scientifically sound, I believe that the authors have omitted some information from the experimental design which is quite important. For instance, there is no mention in the paper of how the treatment with sea water impacted the health of the tea plantations. Additionally, there is a lack of information regarding how the sea water was prepared, which brings up questions of possible microbial contamination from the treatment if the water was not autoclaved.  See below my detailed queries regarding the manuscript.  

Introduction:

Lines 59-60: If the statement implies that there have been several studies indicating the synergistic effect of mycorrhizal fungi and bacteria on plant growth, that the authors should provide more than one reference. Additionally, in my opinion the authors should discuss in more detail the role of PGP microorganisms and some of the mechanisms involved in plant promoting and stress resistance.

Methods:

Lines 92-96: These statements don’t make sense and display inaccurate language. What do the authors mean by “test results”? In addition, the authors should include more detailed descriptions of how the seawater was prepared and administered. What was the pH, EC and hardness of the control water? Additionally, what does EC stand for – please include the definition the first time you use the abbreviation.

Line 98: Why were the samples only collected 5 weeks after the beginning of the experiment? The authors should include the reasoning for this timeline. Did the authors perform a pilot study to determine the optimal sampling time?

Line 116: Either include a reference to the exact method you used, or include a description of the method in the text.

Lines 120-121: Please include the sequences of the primers, or references to these sequences.

Line 135: Please indicate the coverage of the sequencing.

Lines 145-146: Please indicate the version of the databases used to train the classifiers, as this is a crucial detail for reproducibility.

Lines 153-158 : Include references of statistical analyses and packages where appropriate.

Lines 158-162: As highlighted above, please include references to statistical analyses where appropriate. Additionally, the authors should clarify why both EdgeR and LEfSe  are used, as both are differential abundance analysis. Additionally, the authors should describe how the data was normalized before performing these tests, as they both use different normalization methods.

Lines 165-185: Include references of statistical analyses and packages where appropriate.

Results:

Lines 193-195: This is a partly a contradiction and repetition of previous statements regarding the effect of seawater on soils. Do you mean bulk soils? Please clarify in the text. Additionally, are these changes significant?

Lines 285-291: I find it interesting that while ARS demonstrated lower prokaryotic diversity than NRS, it resulted in a more complex network, which would imply greater functional redundancy and resilience. Can the authors explain the reason why this was the case?

Lines 292-296: Please indicate the thresholds used for defining hubs in the methodology.

Line 313: Delete the word “uniquely”.

Discussion:

Lines 334-335: Incorrect wording – since you did not test other abiotic factors in your analysis, you cannot state that the seawater contamination dominated the differences between groups. Rather, you can say that the groups could be separated based on the treatment.

Lines 343-345: This is only the case with the fungal dataset – in the case of prokaryotes, the seawater treatments resulted in significant lower diversity. Please correct.

Line 369: What do the authors mean by “strong vitality”? Please rephrase in scientific terms.

Lines 383-385: Incorrect phrasing – please rewrite.

Lines 418-420: In my opinion, the authors should start this discussion by describing the limitations of FAPROTAX as a predictive tool based on functional inferences from previous literature.

Data availability: As someone who believes in open science, in my opinion the authors should have the raw data and metadata available in a public repository such as NCBI upon publication. Failure to do so invites suspicion regarding the quality of the datasets, and decreases the relevance of the study by preventing reproducibility.

Figure 1: Figures too small. Please increase them

Figure 2: Please include the sample group acronym on the horizontal bar of the heatmaps. In addition, the heatmaps are too small.

Comments on the Quality of English Language

Minor errors on the grammatical structure need to be corrected. 

Reviewer 2 Report

Comments and Suggestions for Authors

The study provides novel insights into the response of tea plant rhizosphere microbiome to seawater stress, an important topic given the increasing threat of coastal salinization. The findings contribute to the understanding of plant-microbe interactions and the role of the rhizosphere in mediating plant adaptation to abiotic stress, which has broader implications for sustainable agriculture. The topic is of high relevance to researchers in the fields of plant-microbiome interactions, abiotic stress, and sustainable agriculture, particularly in coastal regions. The manuscript is well-structured, with clear reporting of, results, and discussion. The figures and tables effectively convey the key findings. The study employs robust molecular techniques and bioinformatics tools, although some details should be improved, as detailed below.

The language used is generally clear, concise, and easy to understand. Occasionally, there are minor grammatical errors, such as subject-verb agreement issues or improper use of articles (a, an, the). Some sentences could be improved for better flow and clarity, but the meaning is still conveyed effectively. There are a few instances of awkward phrasing or word choice that could be refined to improve the readability.

The title could be slightly improved to be more concise and specific. E.g. "Microbial Community Shifts in Tea Plant Rhizosphere under Seawater Stress: Enrichment of Beneficial Taxa"

Introduction

Overall, the introduction is well-structured and provides a clear context for the research question and objectives. However, there are a few areas where the introduction could be further strengthened:

Provide more specific details on the known impacts of seawater stress on soil microbial communities and their potential implications for plant health and productivity.

Highlight any previous studies that have explored the tea plant rhizosphere under abiotic stress conditions, to better situate the current research within the existing knowledge base.

Briefly mention the potential significance and broader implications of understanding the tea plant rhizosphere microbiome's response to seawater stress.

Methods

The research design employed in this study appears to be appropriate and well-suited to address the research objectives. However, some limitations should be highlighted and, at least, discussed in the Discussion Section.

The study was conducted in a specific region (Shandong, China) using seawater-contaminated irrigation water. The generalizability of the findings to other tea-growing regions or different seawater stress scenarios may be limited. In addition, the study presents a snapshot of the microbial community changes at a specific time point. Examining the temporal dynamics of the rhizosphere microbiome under prolonged seawater stress could provide additional insights.

The study also focuses on the descriptive aspects of the microbial community changes, but a deeper investigation into the underlying mechanisms driving these changes could enhance the overall understanding.

The study acknowledges the potential role of tea plant root exudates in shaping the rhizosphere microbiome but does not include any direct analyses of the exudates.

Incorporating root exudate characterization could provide valuable insights into the plant-microbe interactions.

My main concern regarding the Method section is the lack of detailed information regarding the specific parameters and settings used in the data analysis process. Providing additional details about the specific parameters and settings used, such as the quality thresholds applied during read merging and filtering, as well as the specific parameters used in denoising and ASV generation, would enhance the reproducibility and transparency of the analysis. Some questions include:

What chimera filtering was used? What Downstream analyses were performed, and what were the parameters used? How were reads normalized? Which method and which parameters were used? The cut-off should be different for fungal and bacterial samples. Any parameter wrongly applied here can significantly impact the results.

In addition, Greengenes is an outdated database, last updated more than 10 years ago. SILVA is an updated database for bacterial taxonomy. Furthermore, due to intraspecific variability in the ITS region, and sometimes intragenomic variability, ITS sequences must be clustered to approach species-level resolution in community studies. As such, ASV is not recommended.

Provide rarefaction curves for bacterial and fungal amplicon sequence variants. In addition, provide a supplementary table with the number of sequences input for each sample, number of sequences after initial filtering, number of chimeric sequences removed, and the final number of sequences (used for microbial community analyses).

Results

The results are presented in a detailed manner, but the authors could consider adding more concise summaries or highlights at the end of each subsection

Discussion

The discussion is structured into several subsections, which helps organize the key points. However, the transitions between subsections could be strengthened to enhance the overall flow and coherence of the discussion.

The discussion acknowledges some of the limitations of the study, such as the geographical constraints and the lack of temporal dynamics. However, the authors could expand on these limitations (as previously I pointed out) and suggest more specific future research directions to address them.

While the discussion covers several important aspects, it could potentially be streamlined in certain areas to maintain a clearer focus on the keynotes.

Comments on the Quality of English Language

The language used is generally clear, concise, and easy to understand. Occasionally, there are minor grammatical errors, such as subject-verb agreement issues or improper use of articles (a, an, the). Some sentences could be improved for better flow and clarity, but the meaning is still conveyed effectively. There are a few instances of awkward phrasing or word choice that could be refined to improve the readability.

Reviewer 3 Report

Comments and Suggestions for Authors

A paper submitted by Long Yang and co-authors to Microorganisms analyses changes in the soil microbiome of tea plants following long-term seawater application.

The authors have done impressive work that corresponds to the topic of the journal. However, some issues should be resolved to make the article more exciting and useful to readers

1) From the Abstract, Introduction, and the other sections, it is unclear what specific hypothesis the authors planned to test before starting the experiments. It would be good to describe in the Abstract (and Conclusions) whether it has been proven, and in the Introduction to describe in detail on the basis of what literary data it was formulated. The hypothesis testing storyline can be drawn through the Results section and discussed in the Discussion.

2) The authors use many phrases, for example, SBS, ARS, NBS and NRS, which should be deciphered in the description of Table 1, Table 2, Table 3, captions to Fig. 1, Fig. 2, Fig. 3, Fig. 4,

3) Please check the sentence in the line 278

4) Table 2: Modularitya -> Modularity?

5) Please describe in the text of the article what Positive, Negative edges, and Average degrees in Table 2 mean.

6) The Conclusions section is too verbose and vague. Writing out individual conclusions in a list, in separate paragraphs, or in a number is recommended.

Reviewer 4 Report

Comments and Suggestions for Authors The following manuscript is dedicated to the impact of seawater stress on the structural and functional changes in bacterial and fungal communities in the rhizosphere of the tea plant. The text is well-written, Sections are informative, balanced and clear. The majority of Figures should be a little bit improved - their size should be increased. The specific comments are listed below:  

Line 12. "16s and ITS techniques" change to "16S rRNA gene and ITS sequencing". "16S" is slang and it is not correct in the context of microbial profiling as an amplicon is analyzed. Check it throughout the text and change to "16S rRNA gene".

Line 103. The reference "Pisa et al., 2011" is not given as a number. Check references throughout the text.   Lines 120-121. References for primer sequences are needed.   Line 127. Why did authors mix "PCR products with an equal volume of 1X TAE buffer" before the electrophoresis? Did you mean that electrophoresis had been performed in 1X TAE? Actually PCR products should be mixed with the loading buffer (dye). Please, correct.   Lines 201-203. Add the abbreviation expansion for SBS, ARS, NBS, NRS, PH to the legend of Table 1 too. It will facilitate the comprehension of the readers.   Figure 1. Size of the Relative abundance histograms should be larger because it is difficult to look over.   Lines 229, 234. "Candidatus_Koribacter, Candidatus_Solibacter" - "Candidatus" should be written in italic and without underscore. Bacterial and fungal genera names should be written in italic too. Check it throughout the text.   Lines 250-251. "Staphylococcaceae_Staphylococcus" - family name and underscore are not needed.   Figures 2-5. All graphics are too small.   Lines 268-270. New valid phyla names have been adopted according to Oren and Garrity, 2021 (DOI 10.1099/ijsem.0.005056), in particular Proteobacteria = Pseudomonadota, Bacteroidetes = Bacteroidota, Firmicutes = Bacillota and etc. Phyla names mentioned in the text should be checked and corrected according to Oren and Garrity, 2021.   Line 331. "The microbial diversity of tea rhizosphere present more stable under seawater stress" - correct "present" to "presents" or "becomes" or "is". Comments on the Quality of English Language

Minor editing of English language required

Round 2

Reviewer 2 Report

Comments and Suggestions for Authors

The authors have made substantial changes in the revised version following my suggestions and the current version is suitable for publication.

Reviewer 3 Report

Comments and Suggestions for Authors

The manuscript has been significantly improved according to the reviewers' comments. The paper can be published in Microorganisms in its current state

Sincerely,